# Sport Motivation from the Perspective of Health, Institutional Embeddedness and Academic Persistence among Higher Educational Students

**DOI:** 10.3390/ijerph19127423

**Published:** 2022-06-16

**Authors:** Karolina Eszter Kovács, Klára Kovács, Fruzsina Szabó, Beáta Andrea Dan, Zsolt Szakál, Marianna Moravecz, Dániel Szabó, Tímea Olajos, Csilla Csukonyi, Dávid Papp, Balázs Őrsi, Gabriella Pusztai

**Affiliations:** 1Institute of Psychology, University of Debrecen, 4032 Debrecen, Hungary; olajos.timea@arts.unideb.hu (T.O.); csukonyi.csilla@arts.unideb.hu (C.C.); papp.david@arts.unideb.hu (D.P.); orsibalazs@arts.unideb.hu (B.Ő.); 2Institute of Educational Sciences and Cultural Management, University of Debrecen, 4032 Debrecen, Hungary; kovacs.klara@arts.unideb.hu (K.K.); pusztai.gabriella@arts.unideb.hu (G.P.); 3Institute of English and American Studies, University of Debrecen, 4032 Debrecen, Hungary; szabo.fruzsina@arts.unideb.hu; 4Doctoral Program on Educational Sciences, University of Debrecen, 4032 Debrecen, Hungary; dan.beata@arts.unideb.hu; 5Kölcsey Ferenc Teacher Training Institute, Debrecen Reformed Theological University, 4026 Debrecen, Hungary; szakal.zsolt@drhe.hu; 6Physical Education and Sport Science Institute, University of Nyíregyháza, 4400 Nyíregyháza, Hungary; moravecz.marianna@nye.hu (M.M.); szabo.daniel@nye.hu (D.S.)

**Keywords:** sport motivation, persistence, health, relationship network, higher education

## Abstract

Regular physical activity from an early age is an important part of a healthy life because if we incorporate exercise early into our lifestyle, we are more likely to maintain our commitment to sport into adulthood and even throughout our lives. In our research, we used the PERSIST 2019 database, which contains data from students at higher education institutions in Hungary, Slovakia, Romania, Ukraine, and Serbia. We used factor analysis to isolate four sports motivation factors (intrinsic, introjected, extrinsic, and amotivation). Factors influencing the different types were measured using linear regression analysis, involving the variables in four models. The results show that the effects of the sociodemographic variables are significant for gender, country, and mother’s job, especially in terms of intrinsic, introjected, and extrinsic motivation. The role of coping is salient for health awareness factors, with a positive effect on intrinsic motivation and a negative effect on the other types. The impacts of quality of education and support are typically negative, while the positive effect of satisfaction with infrastructure is noteworthy. The effect of persistence in sport on intrinsic and introjected motivation is positive. Frequency of training increases intrinsic motivation, while practical sport embeddedness generates extrinsic motivation. In terms of relationships, a mainly teacher-oriented network within the institution typically has a negative effect on intrinsic motivation, while peer relationships outside the institution typically increase intrinsic and extrinsic sport motivation. Academic persistence has a positive effect on intrinsic motivation and a negative effect on introjected motivation. Our research highlights the complexity of factors influencing sport motivation and the role of coping, which typically remains strong when relationship-related variables are included. In addition, we must emphasise the dominant role of relationship network patterns, which may even reduce commitment to sport.

## 1. Introduction

Social-contextual factors (e.g., coaches’ leadership style, climate) and interpersonal variables (e.g., perceived competence, perceived autonomy, self-regulation) are commonly emphasised in motivational theories and have been found to play a significant role in sporting behaviour [1]. As dropout from sport is often attributed to a lack of motivation and self-regulation skills, it is essential to understand the underlying motivational processes [2]. The role of socioeconomic factors is often highlighted in theories, while the role of peers and institutional embeddedness is an under-researched area.

In terms of motivation, a distinction is often made between intrinsic and extrinsic motivation. Intrinsic motivation is when an athlete pursues sport for the sake of the activity itself because it is enjoyable. This is the strongest and most persistent behaviour, and the performance of the action is rewarding (e.g., playfulness, curiosity, interest, fun). Intrinsically motivated individuals have competence goals related to self-determination, excellence, and success, and when these are achieved, they also become rewards. The situation is different with extrinsic motivation: for the athlete, the pursuit of the activity is not intrinsic but is driven by external factors such as money or other material factors. Extrinsic motivation can also come from praise or discipline, a combination of rewards (such as public recognition, prizes, medals, certificates) and punishments (harder training, withdrawal of previous rewards), but such motivation is short-lived and less effective. In this case, once the extrinsic factor is removed, the athlete’s performance can drop dramatically. Therefore, achieving intrinsic motivation seems more desirable [3,4].

There are, however, different types of extrinsic and intrinsic motivation, which Deci and Ryan ranked along a continuum. They vary according to the motivational state of the athlete. Five types have been identified: *extrinsic regulation* (which involves the need to meet some external requirement); *introjected* (*internalised*) *regulation* (the athlete feels that they should do sport, but the motivation does not become intrinsic; instead the external source is replaced by an internal reason such as conscience); *identified regulation* (when the activity is already the result of an individual decision but is still performed to achieve an extrinsic goal); *integrated motivation* (which refers to a person’s intrinsic orientation or desire but still contains external drives as well); and *classic intrinsic motivation* (when the athlete participates in the sport for their own pleasure and satisfaction). As an additional type, *amotivation* can be separated when there is no intrinsic reason to do sports; it can be defined as the lack or absence of volitional drive to engage in any activity resulting from non-self-determined motivation [5].

In our research, we involve the factors influencing sporting habits following the model of Cairney et al. [6]. They summarise the role of individual and environmental influences in the sporting process in their holistic model, in which sport motivation is considered a primary segment (Figure 1). Motivation, social participation, enjoyment of the activity, and physical competence are circularly involved in the process so that one factor influences the other, which can then interact with each other. These foundations are complemented by additional intrapersonal and environmental segments, the end products of which, physical, mental, and social health, can be mapped to the results of the bio-psycho-social model of health. Physical activity itself is thus determined by a number of factors, and in addition to habitus, physical abilities occupation, and academic aspects play a significant role, as reflected in the form, level and embodiment of the sporting activity performed. Academic performance is therefore modelled as determining the way sporting activity is performed, as well as its underlying determinants, and thus may influence sport motivation [6].

Concerning the sociodemographic factors influencing sporting habits, age must be mentioned, as the proportion of young people taking part in sport declines with age [7,8,9,10,11,12,13,14]. There are also significant gender differences in sport motivation as men are more intrinsically motivated, based on the need for achievement, while women are more extrinsically motivated, which is linked to physical attractiveness and appearance [15,16,17,18,19,20]. There are also significant external differences in the focus of sporting activity: male athletes are characterised by a higher ego orientation than female athletes [21,22,23], while female athletes have a higher task orientation than male athletes [24,25,26]. The role of the parents also must be considered [27,28]. Participation, especially early participation, significantly reduces opportunities for gaining experience and sporting capital, negatively affecting disadvantaged families from a low socioeconomic status [29,30]. The motivation to engage in sport is well captured for children of university-educated parents and children from better-off families. Other environmental factors should also be mentioned. Together, socio-environmental factors can be interpreted as motivational climates [30,31] that are relevant in many sport contexts (e.g., teammates, sport structures).

Concerning intrapersonal factors, the role of health is crucial. Principally, intrinsic motivation is associated with higher self-esteem; better quality of life, emotion regulation, and perceived mental health; greater endurance; better quality of sport-related relationships; and, more fundamentally, improved social relationships [32,33,34,35,36]. Higher levels of intrinsic motivation are linked to higher levels of daily well-being and lower risk of burnout [35]. In addition, the increased use of engagement coping and the decreased use of disengagement coping lead to better health and academic achievement [37]. These associations are true for identified, introjected and extrinsic motivation as well, although they are usually less effective and have a smaller impact but can be preventive against negative psychological correlates such as burnout or depression. On the contrary, amotivation is usually positively associated with psychological distress and burnout [32,33].

Regarding the environmental factors, previous research suggests a link between the university environment (social and academic systems) and the academic success of student–athletes [38,39], which the conceptual model described above [6] confirms. However, the direction of the effect is unclear. Some research suggests a positive relationship between sport participation during university and academic achievement [40]. Other research, however, suggests a negative relationship between sport participation outside the university and academic success [41,42]. As with other subgroups of students, the perceptions and experiences of the campus climate influence the athletes’ feelings of support, belonging and competence. Pusztai et al. [43] highlighted that institutional integration, i.e., strong embeddedness in an institutional peer network, does not enhance the protective role of social relationships. On the contrary, solid and multiplex embeddedness in the peer community significantly increases the likelihood of risky behaviour. Therefore, students’ tight intragenerational social networks may be a threat to the development of a healthy lifestyle, which then can be reflected in their sport participation and motivation.

The academic environment of student athletes is partly influenced by the academic profiles of their non-athlete peers and the institutional educational resources supporting them. This environment puts pressure on athletes to perform academically in addition to their sporting activities. Many athletes find themselves in difficult sporting and academic contexts, but this is particularly challenging for female athletes. At some universities (e.g., in the USA), athletes who underperform academically compared with their peers are at risk of not being able to participate in competitive sport. These pressures can potentially affect the academic success of student–athletes as well as their sporting success. Despite these challenges, some research suggests that female athletes show the same commitment to studies and sport as men [15,44]. As a result, female athletes tend to perform better academically [39].

This research aims to detect the different types of sport motivation following the model of Deci and Ryan [5]. We expected to detect at least four types mentioned in the original model. We also aimed to explore the various intra- and interpersonal and environmental factors influencing students’ sports motivation in testing the following hypotheses:

**Hypothesis** **(H1).***The role of sociodemographic factors is significantly positive concerning all types of motivation except for amotivation (especially in case of gender, parents’ education level and objective and subjective financial situation)*.

**Hypothesis** **(H2).***Coping, trust and satisfaction with institutions significantly increase intrinsic and introjected motivation and decrease extrinsic motivation and amotivation*.

**Hypothesis** **(H3).***Sport-related variables and institutional networks significantly increase intrinsic, introjected and extrinsic motivation and decrease amotivation*.

**Hypothesis** **(H4).***Academic persistence supports intrinsic motivation but hinders introjected and extrinsic motivation*.

We aimed to test our hypotheses based on the validation of an instrument (BREQ-2) [45] that has not been used in Hungary yet. Since our main aim was to examine factors influencing sport motivation, other intra- and interpersonal and environmental factors were treated separately (see Section 2.2). 

## 2. Materials and Methods

### 2.1. Sample

In our research, we used the PERSIST 2019 database. The PERSIST 2019 research was conducted in 2018–2019 among students in higher education institutions in the Northern Great Plain and four cross-border regions (Highlands in Slovakia, Transcarpathia in Ukraine, Vojvodina in Serbia, Transylvania, and Partium in Romania) where Hungarian is the language of instruction (The former territories of Hungary that now belong to Romania are called Partium and Transylvania, those in the Ukraine are referred to as Subcarpathia, and those in Serbia are called Vojvodina. The students participating in our survey come from these territories, so the names of the countries will be used synonymously with the territories listed earlier. However, it is to be noted that our results and findings only apply to the institutions of these territories, and they are not representative of the entire countries. The following institutions were involved into the study: University of Debrecen [Debreceni Egyetem] (*n* = 803), Debrecen Reformed Theological University [Debreceni Református Hittudományi Egyetem] (*n* = 19), University of Nyíregyháza [Nyíregyházi Egyetem] (*n* = 112) (Hungary *n* = 934); Babes-Bolyai University and its outsourced faculties [Babes–Állami Egyetem] (*n* = 173), Partium Christian University [Partiumi Keresztény Egyetem] (*n* = 64), Sapientia Hungarian University of Transylvania [Sapientia Erdélyi Magyar Tudományegyetem] (*n* = 135) (Romania *n* = 647); Ferenc Rakoczi II Trascarpatian Hungarian College of Higher Education [II. Rákóczi Ferenc Kárpátaljai Magyar Főiskola] (*n* = 105), Mukachevo State University [Munkácsi Állami Egyetem] (*n* = 10), Uzhhorod National University [Ungvári Nemzeti Egyetem] (*n* = 74) (Ukraine *n* = 189); Constantine the Philosopher University in Nitra [Konstantin Filozófus Egyetem Nyitra] (*n* = 31), Janos Selye University, Komárno [Selye János Egyetem, Révkomárom] (*n* = 98) (Slovakia *n* = 129); University of Novi Sad, Novi Sad and Subotica [Újvidéki Egyetem, Újvidék és Szabadka] (Serbia *n* = 93)). We conducted a quantitative study in higher education institutions in the eastern region of Hungary, Slovakia, Romania, Ukraine, and Serbia. The sample taken in Hungary (*n* = 1034) is representative of the faculties, the subject matter of the courses and the forms of financing. At the institutions across the borders, we made efforts at probability sampling (*n* = 1165). We contacted the students in groups at their college/university courses, and we asked them the entire range of our questions. Students in the same years (2nd and 3rd) were interviewed. Paper-based questionnaires were completed by students participating in courses specifically designed for this research under the instructor’s supervision. Thus, these students participated as research staff. We paired students from different fields of study to complete the specified numbers of questionnaires.

### 2.2. Instruments

The research aimed to investigate the factors that influence sport motivation, and thus we first examined the subscales of sport motivation. This was based on Markland and Tobin [46], whose method is currently being adapted in Hungary [47]. Based on factor analysis, four factors could be identified: intrinsic motivation, introjected motivation, extrinsic motivation and amotivation (method: maximum likelihood, rotation: varimax, total variance explained: 61.48) (Appendix A, Table A1). Therefore, even though the model fit measures were acceptable (χ^2^ = 2412; df = 146; CFI = 0.907; TLI = 0.891; RMSEA = 0.084; SRMR = 0.078), we were not able to create all five of the original dimensions like in the original questionnaire and theory. To provide a wide picture, supposing that different motivational types have different attributes, we aimed to investigate all of the motivation types to detect the differences between the factors influencing them (Contrary to the original categorisation, we did not identify all five original dimensions (intrinsic motivation, identified motivation, introjected motivation, extrinsic motivation and amotivation) in the Hungarian sample. In this population, we could not separate out the category of identified regulation). Since motivational types vary, we supposed that factors influencing them may be different as well concerning their appearance and effect size. Therefore, these types needed to be investigated separately to be able to visualise the various effects of the different intra- and interpersonal and environmental factors. 

As a subjective health awareness factor, a resilience index was created based on the 10-item Connor-Davidson Resilience Scale (4-point Likert scale, CD-RISC-10). In addition, a trust index was created on which respondents indicated their level of trust on a 4-point Likert scale, in the following individuals and institutions: the management of the institution and faculty, administrators, instructors, group/fellow students, student council, and people who send messages in the electronic system of the institution (Neptun). Satisfaction with the institution was also measured in this category. On a 4-point Likert scale, students indicated their level of satisfaction with various factors such as the professional knowledge of the instructors, the quality of the instructors, the scientific research and talent management activities of the instructors, their level of interest in the courses, the usefulness of the curriculum, the helpfulness of the administrative staff, library facilities, student accommodation, etc. Five factors were created using factor analysis: communication, counselling, and talent management; leisure activities; support; quality of teaching; and infrastructure (method: maximum likelihood, rotation: varimax, total variance explained: 44.054%) (Table A2, Appendix A).

To examine engagement in sport, we created a sport persistence index [48], which refers to successful sporting activities that started in the past and are still ongoing. It integrates elements such as whether the students have won a sports scholarship, received extra points in the admission process for sports performance, are members of a sports club/association and whether they are currently pursuing sports (principal component analysis, total variance explained: 43.57%).

The sport participation variable measures the regularity of participation on a scale of 0 to 100, where 0 means never and 100 means a daily sporting activity. The practical sport embeddedness index was created using principal component analysis based on whether the students have a lecturer or a fellow student with whom they can discuss sport-related topics, whether they regularly use the institutional sports infrastructure and whether they participate in university sport programmes (total variance explained: 43.7%).

For the relationship networks within and outside of the institution, five factors were created by factor analysis (method: maximum likelihood, rotation: varimax, total variance explained: 44.359%), as follows:intergenerational integration, where the student–student relationship type dominates;intergenerational integration, dominated by intra-institutional student–student relationships;intragenerational disintegration, reflecting the dominance of the student’s peer relationships out of the institution;practice-focused mixed peer orientation, which is dominated by on- and off-campus relationships that are mainly oriented towards common leisure activities and sports;culture-focused mixed peer orientation, where both on- and off-campus relationships can be found primarily oriented towards learning, education and knowledge.

### 2.3. Analysis

Factors affecting sport motivation were examined using hierarchical multiple regression analysis. For each motivational factor, the effects of the variables were examined in four models. The first model included demographic background variables (gender, country, level of training, type of settlement, mother’s and father’s education, mother’s and father’s employment status, family and one’s own objective and subjective financial situation). In the second model, components of subjective health awareness were examined (coping, trust and five satisfaction factors: communication, counselling and talent management, leisure time activities, support, quality of teaching, infrastructure). The third model included sport-related persistence and institutional relationship variables (sport persistence, sport participation frequency, practical sport embeddedness, the relationship networks introduced above). Finally, in our fourth model, we investigated the effect of academic persistence. Multicollinearity was also checked by the variance inflation factor (VIF). The mean values of the different motivation types according to the sociodemographic variables are presented in the Appendix A (Table A3).

### 2.4. Ethical Statement

The research was conducted in accordance with the Declaration of Helsinki. The institutional review board of the Institute of Educational Sciences and Cultural Management of the University of Debrecen approved the investigation (04_2021). From the participants, no written consent was necessary. Anonymity was assured.

## 3. Results

### 3.1. Intrinsic Motivation

First, we investigated the factors influencing intrinsic motivation (Table 1). Intrinsic motivation reflects on participating in sport for internal reasons, particularly for pure enjoyment and satisfaction. This type considers pursuing sport a pleasurable activity. Playing a sport means having fun, and it leads to satisfaction. Exercising is a personal need and positive value. 

In our first model, we tested the effects of the demographic background variables. The effect of gender is significant (*p* = 0.023), with men having higher levels of intrinsic motivation. The effect of country is also significant (*p* = 0.044), with higher levels of intrinsic motivation among students studying in Hungary. Maternal education (*p* = 0.056) and employment (*p* = 0.056) also have a positive effect, as children with mothers educated at the tertiary level as well as those whose mothers work also have higher levels of intrinsic motivation. In addition, a trend-level effect was observed for education level (*p* = 0.084), as higher levels of intrinsic motivation were found among those seeking a master’s degree, which is presumably related to age. No significant effect was observed for the other demographic background variables. The first model with an R-square of 0.097 suggests that sociodemographic variables only account for 9.7% of the variance in intrinsic motivation. The adjusted R-square is also too small to be considered. The F-change (from 0 to 5.351) for model 1 is, however, significant (*p* < 0.001).

In our second model, we considered psychological variables related to health awareness. The positive role of coping in sport motivation is evident, as more effective coping leads to higher intrinsic motivation (*p* < 0.001). However, we did not find a significant effect for trust (*p* = 0.830), or for most of the satisfaction components, as only satisfaction with infrastructure showed a significant positive effect (*p* = 0.059). When including psychological variables, the significant effect of gender disappears. A tendency-level positive effect is also observed for country (*p* = 0.069), level of training (*p* = 0.057) and mother’s education (*p* = 0.530). In this model, the R-value increased from 0.312 to 0.366 and the R-square from 0.097 to 0.134. This means that this model can account for only 13.4% of the variance in intrinsic motivation. The F-change (from 5.351 to 3.555) for model 2 is significant (*p* = 0.001).

Our third model focuses on the effects of persistence and embeddedness in sport, as well as intra- and extra-institutional relationships. As expected, persistence in sport has a significant positive effect on intrinsic sport motivation (*p* < 0.001), similar to the effect of sporting frequency (*p* < 0.001), as intrinsic motivation deepens with increasing regularity. As for the relationship patterns, intergenerational integration has a surprisingly significant negative effect on intrinsic motivation (*p* < 0.001), while intragenerational disintegration has the opposite, i.e., a positive, effect (*p* = 0.017). The effect of practical sport embeddedness is not significant (*p* = 0.168). Concerning sport persistence and sport embeddedness, multicollinearity must be mentioned (sport persistence: 9.487; sport embeddedness: 9.352). When including the variables of sport persistence and integration, the effect of the demographic background variables is slightly modified. A significant effect of country (*p* = 0.043) can be observed, but the effect of mother’s education disappears (*p* = 0.082), the effect of mother’s employment is seen at trend level (*p* = 0.086) and the effect of the other background variables is not significant. The new variables remove the significant effect of coping as it remains only at trend level (*p* = 0.094). In the third model, the R-value increased from 0.379 to 0.350 and the R-square from 0.097 to 0.134. This means that this model can account for 35% of the variance in intrinsic motivation. The F-change (from 3.555 to 28.639) for model 3 is significant (*p* < 0.001).

Finally, we measured the role of academic persistence in our fourth model, which was significantly positive (*p* = 0.017). That is, the higher the student’s commitment to their studies, the higher their intrinsic motivation. In the model, the effect of country remains outstanding (*p* = 0.025), the trend effect of the mother’s level of educational is maintained (*p* = 0.090) as in the previous model and the other demographic background variables follow the trends of the previous model. The effect of coping remains trend level and positive (*p* = 0.098), and there is also a trend-level positive effect of satisfaction with leisure activities (*p* = 0.084) and a trend-level negative effect of satisfaction with the quality of education. The effects of sport persistence and sport frequency remain strong and positive, and the negative (*p* < 0.001) and positive (*p* = 0.019) effects of intergenerational integration and intragenerational disintegration are still detectable. In the fourth model, the R-value increased from 0.616 to 0.621 and the R-square from 0.379 to 0.385. This means that this model can account for 38.5% of the variance in intrinsic motivation. The F-change (from 28.639 to 5.635) for model 4 is significant (*p* < 0.001). Based on the results, this model best interprets the factors influencing intrinsic motivation.

### 3.2. Introjected Regulation—More or Less?

Next, we examined factors that influenced introjected motivation (Table 2). This type is not far from intrinsic motivation; however, it contains elements rather related to external motivation. In this case, the athletes participate in the activity due to feeling pressured and avoiding feelings of guilt, restlessness, anxiety or shame. Skipping the activity may also lead to the sense of failure. 

In our first model, we measured the role of demographic background variables. Of the variables, only maternal employment showed a significant positive effect (*p* = 0.020). The effects of gender, country and level of training were negligible, as were the father’s education and employment as well as both family’s and own objective and subjective financial situations. Thus, the role of sociodemographic background is not significant in this case. The first model with an R-square of 0.025 indicates that sociodemographic variables only account for 2.5% of the variance in intrinsic motivation. The adjusted R-square is also too small to be considered (0.005). The F-change (from 0 to 1.252) for model 1 is not significant (*p* = 0.244).

In our second model, we also included psychological variables of health awareness. The effect of coping was trend level and negative in the model (*p* = 0.072), and we found no significant effect for confidence (*p* = 0.927). For satisfaction, however, a significant connection was found. The effect of satisfaction with leisure activities is significantly positive (*p* = 0.017), so the more satisfied one is with leisure opportunities, the higher the level of introjected regulation. However, the effects of satisfaction with support (*p* = 0.074) and with quality of teaching (*p* = 0.021) are significantly negative, so these segments reduce the level of introjected motivation. The inclusion of the psychological variables does not change the effects of the demographic background variables. Only maternal employment remains significant. In this model, the R-value increased from 0.157 to 0.240 and the R-square from 0.025 to 0.057. This means that this model can account for only 5.7% of the variance in introjected motivation. The F-change (from 1.252 to 2.925) for model 2 is significant (*p* = 0.005).

In our third model, we included variables related to sport and relationship networks. Sport persistence showed a significant positive effect (*p* < 0.001), i.e., a significant increase in the level of motivation. No significant association was found for frequency of participation in sport (*p* = 0.732) or practical sport embeddedness (*p* = 0.761). However, concerning sport persistence and sport embeddedness, multicollinearity appears (sport persistence: 9.472; sport embeddedness: 9.350). Relationship patterns also do not show any significant effect, except for the negative trend-level effect of intragenerational disintegration (*p* = 0.078). For the demographic background variables, effects of maternal employment (*p* = 0.090) and gender (*p* = 0.031) are observed, reflecting that the effect of male regulation is lower, as in the previous model. Looking at the psychological variables, coping strengthens to a significant negative effect (*p* = 0.001), but the effect of the satisfaction component disappears, leaving only the trend-level negative effect of satisfaction with support (*p* = 0.067). In the third model, the R-value increased from 0.240 to 0.409 and the R-square from 0.057 to 0.167. This means that this model can account for 16.7% of the variance in introjected motivation which is still low. The F-change (from 2.925to 9.552) for model 3 is significant (*p* < 0.001).

Finally, our fourth model includes academic persistence, which has a significant negative effect (*p* = 0.021); thus, persistence in studies worsens the level of introjected regulation. In our model, the effect of gender becomes significantly negative (*p* = 0.031), the role of maternal employment remains significant (*p* = 0.047), and the other demographic background variables do not show a significant relationship. The significantly negative effect of coping persists (*p* = 0.003), and support (as in the second model) shows a significantly negative effect on satisfaction-related factors (*p* = 0.036). As for the embeddedness variables, the significant positive effect of sport persistence is maintained (*p* < 0.001), but the relationship patterns remain insignificant: as in the previous model, a trend-level negative effect of intragenerational disintegration is observed (*p* = 0.083). In the fourth model, the R-value increased from 0.409 to 0.418 and the R-square from 0.167 to 0.175. This means that this model can account for 17.5% of the variance. The F-change (from 9.552 to 5.350) for model 4 is significant (*p* < 0.021). Based on the results, this model best interprets the factors influencing intrinsic motivation; however, the variance explained is still extremely low.

### 3.3. Segments Influencing Extrinsic Motivation

We also measured the factors that influence extrinsic motivation (Table 3). This type reflects on participating in sports for external reasons. In this respect, concerning material reasons, receiving rewards or avoiding punishment can be mentioned. Concerning interpersonal relationships, the roles of family members, friends, coaches and partners (i.e., significant others) should be emphasised.

The effect of gender is not significant with regard to the demographic background variables (*p* = 0.323). In contrast to intrinsic motivation, the effect of country is not significant (*p* = 0.570). Essentially, the effects of the demographic background variables are not confirmed for extrinsic motivation. Only a trend-level positive effect is observed for mother’s level of education (*p* = 0.062). In the first model with an R-square of 0.017, sociodemographic variables only account for 1.7% of the variance in extrinsic motivation. The adjusted R-square is also too small to be considered (0.003). The F-change (from 0 to 0.859) for model 1 is not significant (*p* = 0.589).

In our second model, we included psychological variables related to health awareness. Coping shows a tendency level (*p* = 0.090), but a negative effect can be found in contrast to intrinsic motivation. That is, those with better coping have lower levels of extrinsic motivation. Confidence also shows no significant effect in this case (*p* = 0.498), but the effect of satisfaction with the support factor is significant (*p* < 0.001) with a negative effect: higher satisfaction with the support factor lowers the level of extrinsic motivation of the student. When psychological variables were included, no change in the effect of sociodemographic variables was detected. In this model, the R-value increased from 0.130 to 0.225 and the R-square from 0.017 to 0.020. The F-change (from 0.859 to 2.990) for model 2 is significant (*p* = 0.004).

Our third model focuses on persistence and embeddedness in sport and the effects of relationship patterns. In this case, persistence in sport also shows the expected result, i.e., a significant negative effect on extrinsic sport motivation (*p* = 0.047). The effects of the practical sport embeddedness variables are also present, as sport frequency tends to have a negative effect (*p* = 0.069), while practical embeddedness shows a significant positive effect (*p* = 0.006). That is, regularity decreases extrinsic motivation, while embeddedness components increase it. Additionally, concerning sport persistence and sport embeddedness, multicollinearity appears in this motivational segment as well (sport persistence: 9.472; sport embeddedness: 9.350). In terms of relationship patterns, a trend-level positive effect is observed for intragenerational integration (*p* = 0.069) and for the mixed practice-focused (*p* = 0.083) as well as culture-focused mixed peer-orientation segments (*p* = 0.083). There are no changes in the demographic variables in the new model, while for the psychological variables, the negative effect of coping is strengthened and becomes significant (*p* = 0.012); the role of satisfaction with support is strengthened as well (*p* = 0.001). At the same time, the trend-level positive effect of the infrastructural agent is also detectable (*p* = 0.094). In the third model, the R-value increased from 0.225 to 0.309 and the R-square from 0.051 to 0.095. This means that this model can account for only 9.5% of the variance in extrinsic motivation. The F-change (from 2.990 to 3.562) for model 3 is significant (*p* < 0.001). Even if the variance explained is still low, concerning our analysis, this model can explain the factors influencing extrinsic motivation the best.

Finally, in our fourth model, we examined the role of academic persistence, which did not show any significant effect (*p* = 0.487). The roles of the demographic background variables in the model remain insignificant. As in the previous model, coping shows a significant negative effect (*p* = 0.017), and the effect of satisfaction with support also remains significantly negative (*p* = 0.001). The effects of trust and of additional satisfaction segments are not significant. The negative effect of sport persistence continues to be observed at trend level (*p* = 0.078), and the positive effect of intragenerational integration (*p* = 0.068) and the positive trend levels of the practice-focused (*p* = 0.088) and culture-focused mixed peer orientation segments (*p* = 0.074) remain. In the fourth model, the characteristics hardly changed (R-value, from 0.310 to 0.621; R-square, 0.095 to 0.96). This model can account for 9.6% of the variance in extrinsic motivation. The F-change (from 3.562 to 0.484) for model 4 is not significant (*p* = 0.487).

### 3.4. Amotivation and Effects

Finally, we also examined the factors that influence amotivation (Table 4). This profile reflects on the lack of any kind of motivation (including the types mentioned before). In this case, a reduction can be seen in the motivation, usually leading to dropping out of the sport. The athlete finds no intrinsic or extrinsic drive to pursue the activity. 

In our first model, country showed a strong effect with regard to the demographic variables (*p* = 0.009), with significantly lower levels of amotivation for students from Hungary. One’s own subjective financial situation shows a significant positive effect (*p* = 0.009): those who have a better subjectively assessed financial situation also have higher levels of amotivation. The effect of gender is not significant in this component, but it does show a trend level (*p* = 0.062). No significant effect was observed for the other demographic background variables. The first model with an R-square of 0.033 indicates that sociodemographic variables only account for 3.3% of the variance in amotivation. The F-change (from 0 to 1.666) for model 1 is not significant (*p* = 0.07).

Our second model focuses on the inclusion of psychological variables related to health awareness. The effect of coping was not detectable in the model (*p* = 0.130), nor was there a significant effect for confidence in this case (*p* = 0.219). In terms of satisfaction, the effects of satisfaction with communication, counselling and talent management were found to be significantly positive (*p* = 0.009), as it was shown that those who were more satisfied with this segment showed higher levels of amotivation. On the other hand, satisfaction with the quality of teaching has a significant negative effect (*p* = 0.048), as it reduces the level of amotivation. Satisfaction with support ha a trend-level positive effect (*p* = 0.077), as higher levels of satisfaction are associated with lower levels of amotivation. The inclusion of the psychological variables slightly modifies the effects of the demographic background variables, as the effect of gender becomes significant (*p* = 0.023) and the effect of country forms become trend level (*p* = 0.075), while the effect of one’s own subjective financial situation remains (*p* = 0.005). In this model, the R-value increased from 0.180 to 0.250 and the R-square from 0.033 to 0.062. This means that this model can account for only 6.2% of the variance in amotivation. The F-change (from 1.666 to 2.675) for model 2 is significant (*p* = 0.01).

Our third model focuses on the impacts of sport and relationships. As expected, we find a significant negative effect for sport persistence (*p* = 0.010), and we also detected a significant negative effect for practical sport embeddedness (*p* = 0.027). Regarding sport persistence and sport embeddedness, multicollinearity appears (sport persistence: 9.252; sport embeddedness: 9.174). The effect of sport frequency is not significant (*p* = 0.176). In terms of relationship patterns, the effects of intergenerational relationship integration (*p* = 0.089) and practice-focused mixed peer orientation (*p* = 0.004) are significantly positive. Intragenerational integration shows a trend-level negative effect (*p* = 0.095). In the model, the effects of the demographic background variables are similar to the previous model, as gender (*p* = 0.015) and one’s own subjective financial situation (*p* = 0.013) continue to have a significant effect, and the effect of country shows a trend-level effect (*p* = 0.088). As for the psychological variables, the effects of coping (*p* = 0.471) and trust (*p* = 0.315) are not detectable. Moreover, only satisfaction with the type of communication maintains a positive effect (*p* = 0.029); the other satisfaction components do not show a significant relationship with amotivation. In the third model, the R-value increased from 0.250 to 0.331 and the R-square from 0.062 to 0.110. This means that this model can account for 11% of the variance in intrinsic motivation. The F-change (from 2.675 to 3.864) for model 3 is significant (*p* < 0.001).

Finally, our fourth model included academic persistence, which showed no significant effect (*p* = 0.107). In the model, the effects of gender (*p* = 0.031) and one’s own subjective financial situation (*p* = 0.019) remain significant, and the effect of country is trend level (*p* = 0.065), with the other demographic background variables showing no significant relationship. The significant positive effect of satisfaction with the communication factor persists (*p* = 0.015), with the other psychological factors showing no relationships. The significant negative effects of sport persistence (*p* = 0.007) and practical sport embeddedness (*p* = 0.025) are also clearly maintained. There are no changes in the types of relationship networks compared with the previous model. There are no changes concerning the relationship network types compared with the previous model. In the fourth model, the R-value increased from 0.331 to 0.337 and the R-square from 0.110 to 0.114. This means that this model can account for 11.4% of the variance in intrinsic motivation. The F-change (from 3.864 to 2.606) for model 4 is not significant (*p* = 0.107). Based on the results, this model best interprets the factors influencing intrinsic motivation.

## 4. Discussion

Motivation is an extremely important factor in sporting activity. There are several factors that can underlie sports that can arise from individuals themselves, their relationships, and the environment, and these can interact positively and negatively to influence the frequency, form and quality of the sporting activity. The frequency of regular sporting activity tends to decline during the university years, which may be due to a variety of factors such as academic pressure, work or lack of motivation.

In our research, we investigated individual factors of sport motivation; intrinsic, introjected and extrinsic motivational regulation; and amotivation, in a sample of higher education students (PERSIST 2019). We were able to create four motivational types, which differs from the original model of Deci and Ryan [5]. We examined the factors influencing the four types of motivation, in each case including independent variables in four models (sociodemographic, psychological, integration and relationship, and academic persistence). Since we supposed that these independent variables would have different effects on the different types of motivation, we analysed the effects of the possible influential factors separately in four regression analyses. This assumption could have been indicated during the analysis since the manifestation of the effect of hypothesised influential factors was diverse. 

In all cases, the demographic background variables provided the first pillar. Based on our results, our first hypothesis is partly confirmed. Our results show that the effect of gender was significant for intrinsic motivation and amotivation and that men were significantly more likely to have higher intrinsic motivation but also higher amotivation. In the case of intrinsic motivation, gender is not significant per se but becomes significant when combined with the psychological, integration and academic variables. In addition, gender has a negative bias: that women are more likely to have introjected regulation. This is in line with previous results that women are more motivated by external drives [17,18,19]. The effect of the country is also detectable, with students in Hungary having higher levels of intrinsic motivation and lower levels of amotivation. Thus, it appears that students in Hungarian institutions are more engaged in sporting activities and are primarily driven by intrinsic motivation. The effects of the mother’s level of education and mother’s employment was also evident for several components, typically positive. For the intrinsic, introjected and extrinsic motivation types, the motivation score was higher among children with mothers who had a tertiary-level education. The effect for amotivation, although negative, is not significant. Previous research has also linked sport motivation to parental education, which provides better opportunities and scope for students to learn about sport from an early age and to engage in regular physical activity as part of their lives in the longer term [17,26,27]. The father’s education level and employment were not significant for any of the motivational factors. Similarly, the effect of training level is not significant. Personal and family objective and subjective financial situation were not particularly important in motivation. The effect of one’s own subjective financial situation is significant for amotivation: those who are subjectively perceived to be in a better financial situation also have a higher amotivation score. The exact background to this needs to be investigated further, but it is possible that those with a more stable financial background are more engaged in other leisure activities and less interested in sport.

Overall, our second hypothesis is partly confirmed. Regarding the variables related to psychological health, the effect of coping was found to be prominent, confirming the results of previous research [10,47]. The effect on intrinsic motivation was positive, while the effects on introjected and extrinsic motivation were significantly negative. That is, coping facilitates intrinsic motivation, while it weakens extrinsic and intrinsic regulation, and no significant effect was found for amotivation. A positive relationship between a strong and resilient personality and intrinsic motivation has been shown in previous research. Higher levels of coping have a positive effect on intrinsic sport motivation through increased self-esteem and intrapersonal efficacy, generally outweighing the effects of extrinsic factors that are less necessary for the individual. Satisfaction with the institution where the student is studying may also have a positive effect on the student’s health and academic performance [49,50]. Factors associated with satisfaction were most evident for introjected regulation. In terms of intrinsic motivation, satisfaction with the infrastructure showed a significant positive effect. Thus, an adequate infrastructural (in this case, sport infrastructure) is essential to increase intrinsic motivation, as students have a wider range of opportunities available. In terms of the introjected regulation, satisfaction with recreational activities predicts a positive effect, while satisfaction with support and quality of teaching predicts a significantly negative effect. Thus, recreational opportunities support introjected motivation, encouraging the student to participate in sport, while support and quality of teaching, which tend to reinforce learning, weakens it. At this level, the question arises: if an athlete must choose between sport and study, which should they choose? Of course, this question needs deeper analysis, but the current findings suggest that the institutional context of adequate education works against sport rather than for it. The same is true for extrinsic motivation. That is, it is not the type of motivation that is important in terms of why a student pursues sport, but rather the quality and type of facilities provided by the institution, as dissatisfaction with these may even reduce the commitment to sport. In addition to the abovementioned reasons, it can also be observed that introjected regulation is not yet equivalent to fully intrinsic, person-induced regulation but rather stems from the environment. It is also important to note that due to the globalization of higher education, teachers became clerks rather than pedagogues due to the increased amount of administrative tasks. These tasks often distract teachers from qualitative support, making it difficult to help students outside of their education tasks. In terms of amotivation, similar segments come to the fore, presumably by a similar analogy.

Our results also confirm our third hypothesis. The effect of persistence in sport was prominent across all models and motivational types, with significant positive effects on intrinsic and introjected motivation and negative effects on extrinsic and amotivation. Thus, sport motivation, which is mostly or partially intrinsic, is increased by a long-term commitment to sport, while extrinsically supported motivation and its absence decrease, maintaining continuing commitment to sport. Although the frequency of sporting activities declines with age, a significant proportion of university students still engage actively and regularly in sport, whether competitively in a university or extra-collegiate sports club or at amateur or hobby level [47]. As with motivation, long-term commitment to sport, that is, the maintenance of the so-called sport personality, is influenced by intrapersonal characteristics and relational–environmental factors. Sport persistence is a determinant of sport motivation and can also be interpreted as its opposite, i.e., sport motivation is one of the pillars of sport persistence [48]. In this context, however, it should be understood in a broader context, as it does not only refer to the commitment to sporting activity but also includes the success of the activity, i.e., the quality factor. The links between sport motivation and sport persistence need to be further explored and investigated in the future, as no research has been conducted in this area and in this region.

The role of sport frequency is similar to that of sport persistence; it stimulates intrinsic motivation and weakens extrinsic motivation, strengthening the results of previous research findings, e.g., [33,43,47,48]. The effect of practical sport embeddedness is the opposite due to the relationship orientation of the variable, including typically extrinsic motivators. In terms of relationship patterns, their role is considered less important for motivational segments. Relationships within the institution (especially with instructors) typically have a negative impact on intrinsic sport motivation [43]. In contrast, peer relationships outside the institution typically increase both intrinsic and extrinsic sport motivation. At the same time, they have a positive effect on amotivation, which is not surprising given that relationships with instructors tend to have a more motivating effect on studies and academic persistence than on sport participation. In the case of amotivation, it can be seen that peer relationships within the university work more against than in favour of sport. As shown for the satisfaction factors, close relationships with instructors have a weakening effect on sport motivation, as these relationships are likely to lead students in a different direction, typically towards the worlds of science and culture. Meanwhile, intragenerational relationships, including peer relationships, provide greater scope for sporting activity, in both individual and peer contexts. However, the problem of multicollinearity, which means the occurrence of high intercorrelations among two or more independent variables in a multiple regression model, more specifically sport persistence and sport embeddedness, must be pointed out. This fact is not unsurprising, as the variables concerned (sport persistence and sport embeddedness) display similar characteristics and are strongly related to sport in the same way as the measured types of motivation.

Finally, our fourth hypothesis is proved to be true. Academic persistence has a positive effect on intrinsic motivation, i.e., commitment to studies increases intrinsic motivation to pursue sports, while its effect on introjected regulation is negative. Fundamentally, the relationships between sport participation and academic performance show an ambivalent pattern, which depends on the level of the student’s participation in sport [18,45,46,47,51]. Competitive and elite athletes have different sport motivational elements such as attitudes towards learning, and the need for study-related knowledge may also differ.

## 5. Conclusions

Overall, sport motivation is influenced by a number of factors such as intra- and interpersonal [17,31,48] as well as environmental and sociodemographic components [17,33,48]. In addition, generally, motivation can be detected as an element of persistence (sport persistence in this case) [49,50,51]. The findings highlight the role of sport motivation as a segment of sport persistence, which should be investigated in further research. Among the individual health awareness components, coping and sport participation are the most prominent factors and are strongly reflected in intrinsic sport motivation as well as in relationship patterns as institutional-level variables. Some researchers also indicate the supportive role of eSports. Our previous research results also indicate that playing e-sport games may have a supportive role in coping, as it can serve as a coping strategy in addition to its community-building and relationship-maintaining effects [52]. The supporting role of commitment to studies has also been observed for intrinsic sport motivation. The results provide a sound basis for further research to explore the interactions and relationships between sport persistence and sport motivation and their interrelationships.

## Figures and Tables

**Figure 1 ijerph-19-07423-f001:**
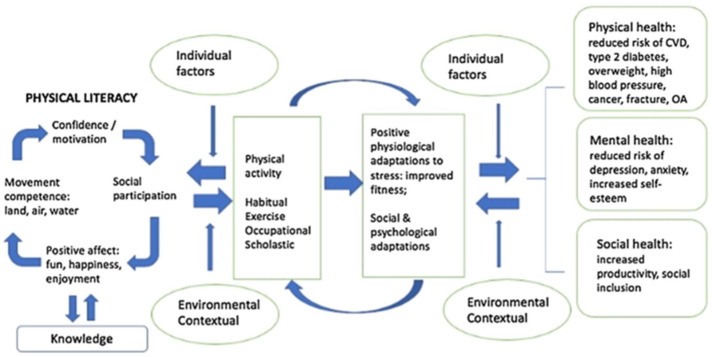
The conceptual model of physical activity and health [6] (p. 373, adapted with permission from Cairney et al., 2019).

**Table 1 ijerph-19-07423-t001:** The factors influencing intrinsic motivation (Exp β) (Source: PERSIST 2019).

	Model 1	Model 2	Model 3	Model 4
**Gender**	0.092 *	0.063	−0.018	−0.005
**Country**	0.090 *	0.084	0.081 *	0.090 *
**Training level**	0.073	0.083	0.040	0.035
**Type of settlement**	0.000	0.083	0.040	0.035
**Mother’s education level**	0.090	0.094 *	0.069	0.067
**Father’s education level**	0.019	0.012	0.031	0.030
**Mother’s employment**	0.081	0.081	0.045	0.045
**Father’s employment**	0.067	0.057	0.060	0.059
**Family’s objective financial situation**	0.076	0.059	0.030	0.035
**Own objective financial situation**	0.009	−0.002	−0.013	−0.016
**Family’s subjective financial situation**	−0.040	−0.061	−0.082 *	−0.074 *
**Own subjective financial situation**	0.062	0.047	0.022	0.015
**Coping**		0.159 *	0.063	0.048
**Trust**		0.010	0.009	−0.001
**Satisfaction (communication, guidance, talent management)**		−0.034	0.039	0.023
**Satisfaction (leisure time opportunities)**		0.042	−0.051	−0.061
**Satisfaction (support)**		0.065	0.054	0.064
**Satisfaction (teaching quality**		−0.057	−0.019	−0.035
**Satisfaction (infrastructure)**		0.075	0.037	0.050
**Sport persistence**			0.397 ***	0.402 ***
**Sporting frequency**			0.363 ***	0.352 **
**Practical sport embeddedness**			−0.138	−0.134
**Intergenerational integration**			−0.161 ***	−0.166 ***
**Intragenerational integration**			0.030	0.029
**Intragenerational disintegration**			0.081 *	0.080 *
**Practice-oriented mixed peer orientation**			−0.050	−0.048
**Culture-oriented mixed peer orientation**			0.015	0.008
**Academic persistence**				0.092 *
**R**	0.312	0.366	0.616	0.621
**R square**	0.097	0.134	0.379	0.385
**Adjusted R square**	0.079	0.106	0.350	0.356

*: *p* < 0.05; **: *p* < 0.01; ***: *p* < 0.001, tendency-level differences (0.05 < *p* < 0.1) are marked with yellow; Gender: 1 = male, 0 = female; Country: 1 = Hungary, 0 = other; Level of education: 1 = MA/MSc, 0 = BA/BSc; Type of settlement: 1 = larger city/county seat/capital, 0 = smaller city/village/farm; Mother’s/Father’s education level: 1 = higher education (at least bachelor’s), 0 = primary or secondary education; Mother’s/Father’s employment: 1 = employed, 0 = not employed; Family’s/Own objective financial status: 1 = above average, 0 = below average; Family’s/Own subjective financial status: 1 = above average, 0 = below average; the other variables are scales.

**Table 2 ijerph-19-07423-t002:** Factors influencing introjected motivation (Exp β) (Source: PERSIST 2019).

	Model 1	Model 2	Model 3	Model 4
**Gender**	−0.014	−0.018	−0.076	−0.091 *
**Country**	−0.068	−0.060	−0.038	−0.047
**Training level**	0.021	0.029	0.003	0.009
**Type of settlement**	0.018	0.013	0.004	−0.004
**Mother’s education level**	−0.070	−0.062	−0.078	−0.076
**Father’s education level**	0.024	0.030	0.037	0.039
**Mother’s employment**	0.102 *	0.096 *	0.082 *	0.082 *
**Father’s employment**	0.012	0.003	0.003	0.004
**Family’s objective financial situation**	−0.005	−0.002	−0.013	−0.020
**Own objective financial situation**	0.074	0.080	0.065	0.069
**Family’s subjective financial situation**	0.054	0.056	0.032	0.025
**Own subjective financial situation**	−0.017	−0.008	−0.006	0.002
**Coping**		−0.079	−0.148 *	−0.132 *
**Trust**		−0.004	−0.009	0.002
**Satisfaction (communication, guidance, talent management)**		0.042	0.065	0.082
**Satisfaction (leisure time opportunities)**		0.100 *	0.055	0.067
**Satisfaction (support)**		−0.075 *	−0.074	−0.085 *
**Satisfaction (teaching quality**		−0.099 *	−0.055	−0.037
**Satisfaction (infrastructure)**		0.027	0.002	−0.013
**Sport persistence**			0.308 ***	0.302 ***
**Sporting frequency**			0.040	0.051
**Practical sport embeddedness**			0.035	0.031
**Intergenerational integration**			−0.007	−0.001
**Intragenerational integration**			−0.030	−0.029
**Intragenerational disintegration**			−0.069	−0.068
**Practice-oriented mixed peer orientation**			0.050	0.047
**Culture-oriented mixed peer orientation**			0.008	0.017
**Academic persistence**				−0.103 *
**R**	0.157	0.240	0.409	0.418
**R square**	0.025	0.057	0.167	0.175
**Adjusted R square**	0.005	0.027	0.128	0.135

*: *p* < 0.05; ***: *p* < 0.001, tendency-level differences (0.05 < *p* < 0.1) are marked with yellow; Gender: 1 = male, 0 = female; Country: 1 = Hungary, 0 = other; Level of education: 1 = MA/MSc, 0 = BA/BSc; Type of settlement: 1 = larger city/county seat/capital, 0 = smaller city/village/farm; Mother’s/Father’s education level: 1 = higher education (at least bachelor), 0 = primary or secondary education; Mother’s/Father’s employment: 1 = employed, 0 = not employed; Family’s/Own objective financial status: 1 = above average, 0 = below average; Family’s/Own subjective financial status: 1 = above average, 0 = below average; the other variables are scales.

**Table 3 ijerph-19-07423-t003:** Factors influencing extrinsic motivation (Exp β) (Source: PERSIST 2019).

	Model 1	Model 2	Model 3	Model 4
**Gender**	0.041	0.050	0.078	0.073
**Country**	−0.026	0.006	0.020	0.017
**Training level**	0.014	0.020	0.015	0.017
**Type of settlement**	0.050	0.053	0.041	0.038
**Mother’s education level**	0.091	0.092	0.085	0.086
**Father’s education level**	0.013	0.028	0.050	0.050
**Mother’s employment**	−0.040	−0.045	−0.043	−0.043
**Father’s employment**	0.035	0.023	0.015	0.015
**Family’s objective financial situation**	−0.026	−0.019	−0.020	−0.022
**Own objective financial situation**	0.015	0.018	0.036	0.038
**Family’s subjective financial situation**	−0.046	−0.035	−0.032	−0.034
**Own subjective financial situation**	0.023	0.041	0.044	0.047
**Coping**		−0.075	−0.114 *	−0.109 *
**Trust**		0.032	0.053	0.056
**Satisfaction (communication, guidance, talent management)**		0.066	0.039	0.044
**Satisfaction (leisure time opportunities)**		−0.031	−0.043	−0.040
**Satisfaction (support)**		−0.150 ***	−0.139 **	−0.142 **
**Satisfaction (teaching quality**		0.036	0.033	0.038
**Satisfaction (infrastructure)**		0.061	0.069	0.064
**Sport persistence**			−0.081 *	−0.083 *
**Sporting frequency**			−0.221 *	−0.218 *
**Practical sport embeddedness**			0.331 **	0.329 **
**Intergenerational integration**			−0.068	−0.066
**Intragenerational integration**			0.079	0.080
**Intragenerational disintegration**			−0.015	−0.014
**Practice-oriented mixed peer orientation**			0.077	0.076
**Culture-oriented mixed peer orientation**			0.072	0.075
**Academic persistence**				−0.033
**R**	0.130	0.225	0.309	0.310
**R square**	0.017	0.051	0.095	0.096
**Adjusted R square**	0.003	0.020	0.053	0.052

*: *p* < 0.05; **: *p* < 0.01; ***: *p* < 0.001, tendency-level differences (0.05 < *p* < 0.1) are marked with yellow; Gender: 1 = male, 0 = female; Country: 1 = Hungary, 0 = other; Level of education: 1 = MA/MSc, 0 = BA/BSc; Type of settlement: 1 = larger city/county seat/capital, 0 = smaller city/village/farm; Mother’s/Father’s education level: 1 = higher education (at least bachelor), 0 = primary or secondary education; Mother’s/Father’s employment: 1 = employed, 0 = not employed; Family’s/Own objective financial status: 1 = above average, 0 = below average; Family’s/Own subjective financial status: 1 = above average, 0 = below average; the other variables are scales.

**Table 4 ijerph-19-07423-t004:** Factors influencing amotivation (Exp β) (Source: PERSIST 2019).

	Model 1	Model 2	Model 3	Model 4
**Gender**	0.078	0.096 *	0.105 *	0.094 *
**Country**	−0.121 *	−0.085	−0.082	−0.089
**Training level**	0.021	0.028	0.039	0.043
**Type of settlement**	−0.009	−0.004	0.005	−0.001
**Mother’s education level**	−0.027	−0.034	−0.022	−0.021
**Father’s education level**	0.018	0.023	0.004	0.006
**Mother’s employment**	0.048	0.055	0.063	0.064
**Father’s employment**	−0.016	−0.018	−0.016	−0.015
**Family’s objective financial situation**	0.062	0.064	0.070	0.066
**Own objective financial situation**	−0.047	−0.045	−0.052	−0.050
**Family’s subjective financial situation**	0.051	0.048	0.052	0.047
**Own subjective financial situation**	−0.108 **	−0.117 **	−0.102 *	−0.096 *
**Coping**		−0.066	−0.032	−0.020
**Trust**		0.057	0.046	0.055
**Satisfaction (communication, guidance, talent management)**		0.120 *	0.101 *	0.114 *
**Satisfaction (leisure time opportunities)**		−0.038	−0.002	0.007
**Satisfaction (support)**		0.074	0.068	0.060
**Satisfaction (teaching quality**		−0.085 *	−0.072	−0.059
**Satisfaction (infrastructure)**		−0.028	−0.024	−0.035
**Sport persistence**			−0.121 **	−0.125 **
**Sporting frequency**			0.163	0.171
**Practical sport embeddedness**			−0.266 *	−0.269 *
**Intergenerational integration**			0.094 *	0.098 *
**Intragenerational integration**			−0.074	−0.073
**Intragenerational disintegration**			−0.065	−0.064
**Practice-oriented mixed peer orientation**			0.128 **	0.126 **
**Culture-oriented mixed peer orientation**			−0.028	−0.022
**Academic persistence**				−0.075

*: *p* < 0.05; **: *p* < 0.01, tendency-level differences (0.05 < *p* < 0.1) are marked with yellow; Gender: 1 = male, 0 = female; Country: 1 = Hungary, 0 = other; Level of education: 1 = MA/MSc, 0 = BA/BSc; Type of settlement: 1 = larger city/county seat/capital, 0 = smaller city/village/farm; Mother’s/Father’s education level: 1 = higher education (at least Bachelor), 0 = primary or secondary education; Mother’s/Father’s employment: 1 = employed, 0 = not employed; Family’s/Own objective financial status: 1 = above average, 0 = below average; Family’s/Own subjective financial status: 1 = above average, 0 = below average; the other variables are scales.

## Data Availability

Data are available only on request due to ethical restrictions.

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
