# Peer review of "Sport Motivation from the Perspective of Health, Institutional Embeddedness and Academic Persistence among Higher Educational Students"

_ijerph, 2022, doi:10.3390/ijerph19127423_

Round 1
Reviewer 1 Report
This is a revised submission. The authors have sufficiently addressed the concerns/feedback.
Author Response
Thank you very much for the evaluation of the reviewer.
Reviewer 2 Report
I thank the authors for their revision, which have largely improved the manuscript and make the aim of the research more clear.
I have a major concern about the authors' understanding of the self-determination theory, which is a major flaw for the present article: self determination theory does not describe stages for change and is not a model, there are regulations of motivation, acting togheter to support behavior change and maintenance, which means these regulations can not be considered separately, unless their is a presentation of the whole continuum. This is illustrated in several manuscript where a self determined motivation index is calculated with the different regulations. While I think this is an original approach to look at each regulation separetaly (even if one was not measured), this needs to be considered in this paper more strongly.
1) The chapter describing the self-determination theory is very approximative, as there are not only a distinction between extrinsic and intrinsic motivation, amotivation is not considered here. Second, this is a continuum and not stages, where individual adopt and maintain behavior using the combination of different motivation, where introjected motivation could play a role in fostering some behavior at short term. On line 70, this is not a scale, this is a continuum, and there have been different scales created using self determination theory. Moreover, there are not phase, but regulation of motivation. Integrated regulation is missing in authors theoretical part, where this concept is also part of the continuum. This chapter should be modified in accordance
2. The introduction should also provide more insights on why basic need satisfaction and motivational climate were not considered as the classic sequence in self-determination theory
3. The method should justify why the authors did not measured integrated motivation
4. The authors should justify why they choose to analyse each motivation regulation separately.
5. The authors should present the motivation profiles of participants in the results and discuss them, before looking at each one.
6. A final model of which variable influence which regulation could be tested, showing the different relationship would be more elegant and help to summarise this research.
Author Response
Thank you very much for the suggestions of the reviewer. We carried out the following modifications:
- ‘The chapter describing the self-determination theory is very approximative, as there are not only a distinction between extrinsic and intrinsic motivation, amotivation is not considered here.’ – We tried to visualise the differences between these types.
- ‘Second, this is a continuum and not stages, where individual adopt and maintain behavior using the combination of different motivation, where introjected motivation could play a role in fostering some behavior at short term. On line 70, this is not a scale, this is a continuum, and there have been different scales created using self determination theory. Moreover, there are not phase, but regulation of motivation.’ – We modified the terminology based on the suggestions and corrections.
- ‘Integrated regulation is missing in authors theoretical part, where this concept is also part of the continuum.’ – The introduction is completed with a brief introduction of integrated regulation.
- ‘The introduction should also provide more insights on why basic need satisfaction and motivational climate were not considered as the classic sequence in self-determination theory’ – We added a short explanation on p. 4.
- ‘The method should justify why the authors did not measured integrated motivation’. - We added a short explanation on p. 5.
- ‘The authors should justify why they choose to analyse each motivation regulation separately.’ - We added a short explanation on p. 5.
- ‘The authors should present the motivation profiles of participants in the results and discuss them, before looking at each one.’ - We introduced the profiled in the relevant subchapters.
- ‘A final model of which variable influence which regulation could be tested, showing the different relationship would be more elegant and help to summarise this research. - Since the independent variables have different impact on the different motivational types, it is not worth analysing motivation together (unseparatedly). Therefore, we think it is not worth conducting a further regression analysis.
Reviewer 3 Report
.
Author Response

(The authors gave the same response as above.)

Round 2
Reviewer 2 Report
Authors have answered to my comments.
This manuscript is a resubmission of an earlier submission. The following is a list of the peer review reports and author responses from that submission.
Round 1
Reviewer 1 Report
The submitted manuscript examines the role of key variables that influence sport motivation in higher education students. The authors of this work made use of a database with massive information and performed factor analysis to identify four dominant factors. In each factor, the authors then conducted linear regressions to assess the significance of several different variables, such as demographic levels, coping, and persistence. The authors then provided their interpretation of these statistical results in the Discussion (or Summary?!).
The Methods and Results sections deserve more attention:
1) The authors presented different models in linear regression analyses, this is more of a hierarchical regression (adding a set of predictors one by one in different models). If what the authors intended to do was hierarchical regression analyses, then the authors shall evaluate which model serves the best for the dataset (e.g. computing improvement in variance explained or R-squared). The authors may refer to an article, for instance, by the same journal below:
https://www.mdpi.com/1660-4601/18/15/8006
Providing some comments on the best model in the regression analyses will also aid interpretation. If not, there seems no purpose to have multiple models in regression.
2) It will be good if the authors check the multicollinearity (e.g. using VIF) to see if different predictors are correlated. In practice, high collinearity may influence the interpretation.
3) The last 3 paragraphs in the Summary section must contain proper citations if the statements are taken from prior studies.
4) Minor: perhaps it is more of a different custom, but the standardized way is to present the decimal numbers in the table using a dot (e.g. ,002 shall be 0.002).
Reviewer 2 Report
The present articles is identifying predictors of different types of motivation for sport practice, which is interesting aim.
The paper does not entail a clear objective and hypothesis, limiting the understanding of the aim. Moreover, the motivation continuum is not considered, as not all types of motivation are considered and there is no reason to consider each type of motivation separately rather than as part as a whole continuum.
The introduction is poorly structured and the presented model does not illustrate the variables investigated in the study. The rationale for the study and for the choice made is also not convincing.
In the method section, no ethical approval is mentionned, data analysis is missing, as well as why each of the type of motivation was considered, as well as why to choose the different determinants.
The results presented does not entail a presentation of the results of the scores of the motivation and the regression with a single model including a lot of predictor looks more like fishing results, rather than having a rationale for choosing the variables.
Discussion should be reworked accordingly to sort out a key message for the present study.
Reviewer 3 Report
It can be better explained why the sample is considered representative.